Retrospective study on the clinical outcomes and characteristics of acute myeloid leukemia: different outcomes in the same risk group

Meng Fanqiao
Xiang Maoyuan
Xie Huan
Liu Yu
Qi Yan
Zeng Dongfeng zengdf@tmmu.edu.cn
Department of Hematology, Daping Hospital, The Third Military Medical University , Chongqing , China
Anson Lesley
Electronic publication date: 2025 Dec 5
Publication date: 2025
Volume: 13
Electronic Location ID: e20436
Received 2025 Apr 28; Accepted 2025 Oct 30
Copyright: ©2025 Meng et al.
Copyright year: 2025
Copyright holder: Meng et al.
License: This is an open access article distributed under the terms of the Creative Commons Attribution License, which permits unrestricted use, distribution, reproduction and adaptation in any medium and for any purpose provided that it is properly attributed. For attribution, the original author(s), title, publication source (PeerJ) and either DOI or URL of the article must be cited.
License URL: https://creativecommons.org/licenses/by/4.0/

Keywords: Acute myeloid leukemia, European Leukemia Network, Prognostic stratification, Genetic heterogeneity

Funding: The Army Medical Center of PLA Talent Innovation Ability Training Program ZXYZZKY07 Chongqing Natural Science Foundation CSTB2024NSCQ-KJFZMSX0082 This study was supported by grants from the Army Medical Center of PLA Talent Innovation Ability Training Program (ZXYZZKY07) and Chongqing Natural Science Foundation (CSTB2024NSCQ-KJFZMSX0082). The funders had no role in study design, data collection and analysis, decision to publish, or preparation of the manuscript.

==============================
Background

Acute myeloid leukemia (AML) remains a prognostically heterogeneous malignancy despite advances in molecular risk stratification. While the 2022 European Leukemia Network (ELN) guidelines refine risk classification, their accuracy in predicting survival outcomes across genetic requires validation.

Methods

We conducted a retrospective analysis of 154 newly diagnosed AML patients at Daping Hospital, integrating next-generation sequencing (NGS)-based genetic profiling, 2022 ELN risk classification, and clinical outcomes.

Results

The most frequent mutations were FLT3 (26.6%), DNMT3A (21.4%), NPM1 (18.2%), CEBPA (17.5%), and TET2 (15.6%). Median overall survival (OS) and progression-free survival (PFS) were 22.9 months and 14.1 months, respectively. Hematopoietic stem cell transplantation (HSCT), female sex, age < 60 years, and normal karyotype emerged as favorable prognostic factors. No significant differences were observed between allogeneic (allo-HSCT) and autologous HSCT (ASCT). Idarubicin, cytarabine, etoposide (IA ± E) chemotherapy yielded superior survival, while azacitidine+venetoclax (AZA+VEN) regimens underperformed. Conversely, TP53 and KIT mutations correlated with inferior survival, while NPM1, CEBPA mutations predicted longer survival. Notably, significant survival heterogeneity existed within 2022 ELN risk groups, particularly among patients with FLT3, CEBPA, or TET2 mutations.

Conclusions

The ELN risk classification demonstrate limitations in prognostication, particularly for patients with FLT3, CEBPA, or TET2 mutations. Our findings highlight the necessity for refined risk models incorporating additional molecular markers (KIT) and mutation interactions to enhance personalized prognostication. Gene coexistence is also a factor that needs to be considered when determining patient prognosis.

Trial registration

The study was registered on the Chinese clinical trial registry (ChiCTR) platform (No. ChiCTR2500096484).

Introduction

Acute myeloid leukemia (AML) is a molecularly heterogeneous clonal disorder characterized by dysregulated myeloid differentiation and dismal survival rates in high-risk subgroups (Kantarjian et al., 2021). Contemporary risk stratification, guided by the National Comprehensive Cancer Network (NCCN) and European Leukemia Network (ELN) criteria, integrates cytogenetic and molecular abnormalities to inform therapeutic decisions (Lachowiez et al., 2023; Pollyea et al., 2023). The prognostic significance of these classifications underscores their critical role in optimizing AML management.

The 2022 ELN guidelines represent a paradigm shift in AML risk classification, revising key criteria from the 2017 version. Notably, FLT3-ITD mutations—previously stratified based on allelic ratio—are now uniformly classified as intermediate-risk when occurring without adverse genetic features. This reclassification primarily addresses methodological challenges in standardizing allele ratio assessment, the emerging role of measurable residual disease (MRD) dynamics in NPM1-mutated AML, and the therapeutic impact of targeted kinase inhibitors (Falini & Dillon, 2024). In addition, FLT3-ITD MRD analysis, which is based on next-generation sequencing (NGS), has improved the ability to predict survival outcomes (Ediriwickrema, Gentles & Majeti, 2023; Peroni et al., 2023). Mutations in AML with mutations in nine myelodysplasia-related (AML-MR) genes (ASXL1, BCOR, EZH2, RUNX1, SF3B1, SRSF2, STAG2, U2AF1, or ZRSR2) are designated as high-risk, independent of other cytogenetic abnormalities. CEBPA biallelic or monoallelic mutations within the bZIP domain are classified as favorable, irrespective of their allelic status. Additionally, hyperdiploid karyotypes with multiple trisomies are no longer considered complex or high-risk, reflecting advances in understanding their prognostic implications (Bouligny, Maher & Grant, 2023; Lachowiez et al., 2023).

Despite these updates, the 2022 ELN framework faces unresolved limitations (Lo et al., 2023). The combination of MRD and risk stratification can guide the treatment of AML (Cloos, Ngai & Heuser, 2023). In addition, studies have shown that age should also be considered in the risk stratification of AML patients (Pogosova-Agadjanyan et al., 2020). Clinical and molecular heterogeneity persist within risk groups, particularly among patients with “normal” karyotypes or co-occurring mutations (Dohner, Weisdorf & Bloomfield, 2015; Mrozek et al., 2012). For instance, patients classified as favorable or intermediate-risk by ELN 2022 exhibit marked survival variability, underscoring the inadequacy of current stratification criteria. Furthermore, NPM1 and CEBPA mutated AML, while generally sensitive to induction chemotherapy, demonstrate inconsistent long-term outcomes due to high relapse rates (Shi et al., 2024; Su et al., 2022; Suzuki et al., 2005; Wang et al., 2020). The integration of novel therapies—such as venetoclax (VEN)-based regimens and hematopoietic stem cell transplantation (HSCT)—has improved survival even in historically high-risk cohorts, yet these advances challenge traditional risk models (Socie, 2022). AML-RC are classified into an adverse risk group, but the prognosis for these patients varies.

However, persistent outcome disparities within risk strata highlight unmet needs in prognostic precision. This study conducts a comprehensive analysis of clinical characteristics and survival outcomes in newly diagnosed AML patients, addressing unresolved controversies in current risk stratification.

Materials & Methods

Study design and participants

The incidence rate of AML in China is approximately 1.62 per 100,000 people. The data is derived from the “Chinese Disease Database”. we chose an incidence rate of 2 per 100,000 for calculating the required sample size. The sample size calculation formula is as follows: za22P1−Pδ2,α is set at 0.05 (generally, a difference less than 0.05 is considered statistically significant), Zα/2 = 1.96, P is 0.002%, and δ is 0.01. The calculated sample size is 77 cases. We collected 154 cases, and the sample size meets the requirements of this retrospective study. In this study, we used the FAB classification for AML. We conducted a retrospective cohort study of 154 consecutive AML patients (excluding AML-M3) diagnosed between January 2018 and February 2025 in the Department of Hematology, Daping Hospital, Third Military Medical University (Army Medical University). Diagnosis followed 2016 World Health Organization (WHO) criteria, with risk stratification per ELN 2022. Clinical information was retrieved from the electronic medical records.

Endpoints and assessments

Overall survival (OS) was defined as the time from the first day of diagnosis to death or the last follow-up. Progression-free survival (PFS) was measured from the start of therapy to the date of treatment failure, progression, or death from any cause at the last follow-up.

NGS and karyotype analysis

Based on international authoritative guidelines (such as ELN, NCCN) and the Leukemia Expert Committee of the Chinese Society of Clinical Oncology (CSCO), the laboratory of our department has detected 56 genes closely related to myeloid tumors for AML diagnosis. The clinical grading interpretation of the filtered gene mutation sites is in accordance with the Li et al. (2017). Using public databases such as dbSNP, 1,000 Genomes, gnomAD, COSMIC, ClinVar, and laboratory self-built hematological tumor database, the variant data is filtered, screened, and graded to obtain the analysis results. Bone marrow (BM) samples underwent NGS targeting 56 myeloid-related genes, as showed in Table S1 (average depth: 2,000 ×). These genes can be divided into eight categories according to their functions (Table S2). Cytogenetics plays a significant role in AML, influencing aspects such as onset, risk stratification, treatment strategy selection, and survival prognosis. The ELN score classifies AML patients into low-risk, intermediate-risk, and high-risk groups by integrating information on gene mutations, chromosomal abnormalities, and molecular markers, thereby optimizing treatment strategies. G-banding analysis of metaphase spreads (≥20 cells) classified karyotypes per the International System for Human Cytogenetic Nomenclature (ISCN) (Simons, Shaffer & Hastings, 2013).

Study approval and informed consent

The study protocol was conducted by the principles of the Declaration of Helsinki and approved by the Ethics Committee Review Board of the Third Hospital of Army Medical University (2025-09). The study was registered on the Chinese clinical trial registry (ChiCTR) platform (No. ChiCTR2500096484). Written informed consent was obtained from all participants or their legally authorized representatives prior to enrollment in the study.

Statistical analysis

Continuous variables were reported as medians (ranges) and categorical variables as frequencies (%). Survival analyses utilized Kaplan–Meier estimates with log-rank testing. Cox proportional hazards models identified prognostic factors (two-sided P < 0.05). Analyses were performed in R v4.2.2 and GraphPad Prism 9.0.

Results

Baseline patient characteristics

The baseline characteristics of the patients are summarized in Table 1. Of 154 patients (median age: 52 years, range: 15–86), 44.8% were male. Common AML subtypes were AML-M2 (48.1%) and AML-M5 (31.8%). There were three patients with mixed-phenotype leukemia (MPAL) and seven patients with AML-RC. Median white blood cell (WBC) count, hemoglobin, and platelets were 8.79  ×  109/L,78.0 g/L, and 35.0  ×  109/L, respectively. Fifty-four percent had a normal karyotype. A total of 67 patients underwent HSCT, with 54 receiving allogeneic HSCT (allo-HSCT) and 13 receiving autologous HSCT (ASCT). Induction therapies included 7+3 regimens (40.9%), azacytidine+venetoclax (AZA+VEN) (15.6%), and others.

Table 1 Baseline characteristics of AML patients at enrollment.

Characteristics	Value	
Male, n (%)	69 (44.8%)	
Age, M (range) years	52 (15–86)	
Age ≥ 60 years	45 (29.2%)	
Laboratory, median (range)		
WBC, M (range) ×109/L	8.79 (0.62–317.53)	
Hb, M (range) × g/L	78 (27–148)	
PLT, M (range) ×109/L	35 (4–366)	
BM blast (%), median (range)	59.5 (17.50–98.50)	
PB blast (%), median (range)	42.0 (0.0–97.0)	
LDH, median (range) U/L	371 (100.4–5,077.9)	
FAB classification, n (%)		
M0	0 (0.0%)	
M1	10 (6.5%)	
M2	74 (48.1%)	
M4	10 (6.5%)	
M5	49 (31.8%)	
M6	1 (0.6%)	
M7	0 (0.0%)	
MPAL	3 (1.9%)	
AML-MR	7 (4.5%)	
Induction chemotherapy		
IA	16 (10.4%)	
IAE	8 (5.2%)	
DA	47 (30.5%)	
DAE	30 (19.5%)	
CAG	5 (3.2%)	
HA	7 (4.5%)	
AZA+VEN	24 (15.6%)	
Others	17 (11.0%)	
Cytogenetic karyotype		
Normal, n (%)	84 (54.5%)	
HSCT	67 (43.5%)	
Allo-HSCT	54 (35.1%)	
ASCT	13 (8.4%)	
Notes.

WBC white blood cell

HB hemoglobin

PLT hemoglobin

BM bone marrow

PB peripheral blood

BM% proportion of BM blasts

PB% proportion of PB blasts

LDH lactate dehydrogenase

IA idarubicin, cytarabine

IAE idarubicin, cytarabine, etoposide

DA daunorubicin, cytarabine

DAE daunorubicin, cytarabine, etoposide

CAG cytarabine, aclarubicin and granulocyte colony-stimulating factor

HA homoharringtonine, cytarabine

AZA azacitidine

VEN venetoclax

HSCT hematopoietic stem cell transplantation

Allo-HSCT allogeneic hematopoietic stem cell transplantation

ASCT autologous hematopoietic stem cell transplantation

Cytogenetic analysis

Cytogenetic analysis was available for 151 patients. Normal karyotypes were observed in 84 patients (54.5%), and chromosome karyotype analysis failed in three patients (1.9%). Solitary abnormalities were detected in 40 (26%) patients, and two abnormalities were detected in 12 (7.8%) patients. Complex karyotypes were observed in 22 (14.3%) patients, and karyotypes with ≥ 3 abnormalities were detected in 1 (0.6%) patient. According to 2022 ELN (only chromosome analysis), 5.8%, 76.6%, and 15.6% of patients are categorized as favorable, intermediate, and adverse risk, respectively. The cytogenetic data are summarized in Table 2.

Table 2 Cytogenetic analysis in AML patients.

Karyotype at diagnosis	n (%)	
Normal karyotype	84 (54.5%)	
inv (16)	4 (2.6)	
t (16; 16)	1 (0.6%)	
t (8; 21)	13	
Single abnormality	40 (26.0%)	
Two abnormalities	12 (7.8%)	
≥ 3 abnormalities	1 (0.6%)	
Complex karyotype	22 (14.3%)	
ND	3 (1.9%)	
Cytogenetic risk (2022 ELN)		
Favorable	9 (5.8%)	
Intermediate	118 (76.6%)	
Poor	24 (15.6%)	
Notes.

ND not detected

Genomic profiling results

NGS was performed on 154 patients, covering 56 genes. NGS detected mutations in 94.2% of patients. Complex variations involving more than three gene mutations were observed in 56.5% (87/154). The genetic landscapes of these patients are shown in Fig. S1. Of the 56 genes analyzed, six were not detected: CDC25C, ETNK1, GNB1, PIGA, PPM1D, and UBA1. The most frequently mutated genes were FLT3 (26.62%), DNMT3A (21.43%), NPM1 (18.18%), CEBPA (17.53%), TET2 (15.58%), ASXL1 (14.29%), TP53 (13.64%), NRAS (13.64%), and WT1 (12.34%), as illustrated in Fig. 1A. The comutation and exlusivity patterns were examined (Fig. 1B). FLT3 co-occurred with NPM1 and DNMT3A but was mutually exclusive with TP53 and CEBPA. TP53 mutations were associated with DDX41 but excluded NRAS, WT1, FLT3, and NPM1. CEBPA was strongly associated with WT1, but mutually exclusive to both TP53 and FLT3 mutations.

Figure 1 Gene mutation landscape in AML.

(A) Bar graph showing the frequency of mutations in AML. (B) Co-mutations and exclusivity patterns of AML. Blue: Exclusivity relationship; Red: co-occurring relationship. Higher color intensity indicates a stronger association.

ELN risk reclassification

Risk stratification according to both the 2017 and 2022 ELN is shown in Fig. S2. Compared to 2017 ELN, the 2022 update reduced the favorable group (22.7% to 18.1%) and increased the adverse group (42.8% to 47.4%), with minimal change in the intermediate group. Twenty-four patients (15.6%) had discordant risk classifications between the two systems.

Survival

Survival data were available for 146 patients. Median follow-up was 16.8 months (range: 0.5–57.4). At the last follow-up, a total of 79 (54.1%) patients had died. Median OS and PFS were 22.9 months and 14.07 months, respectively. Three-year OS and PFS rates were 41% and 36% (Figs. 2A–2B).

Figure 2 OS and PFS of all patients, based on 2017 and 2022 ELN classifications.

(A–B) OS and PFS of all patients. (C–D) OS and PFS based on ELN 2022 criteria in all patients. (E–F) OS and PFS based on ELN 2017 criteria in all patients.

Impact of ELN stratification on survival

On the basis of the 2022 ELN and 2017 ELN risk stratification models, the median OS (16.9 months vs 16.67 months) and PFS (9.8 months vs 9.8 months) results in the intermediate and adverse risk groups are basically the same, whereas, for the favorable risk groups, the median OS (24.13 months vs 26.33 months) and PFS (12.4 months vs 19.1 months) of the 2022 ELN stratification model are both shorter than those of the 2017 ELN stratification model (Figs. 2C–2F).

Survival with HSCT

Of the 146 patients under follow-up, 67 underwent HSCT, which significantly improved survival outcomes. HSCT significantly improved OS (not reached vs. 11.4 months, P < 0.0001; Fig. 3A) and PFS (not reached vs. 9.3 months, P < 0.0001; Fig. 3B). Our transplant group consisted of 54 allo-HSCT and 13 ASCT. To determine whether the type of transplantation influenced survival, we conducted a further analysis. No significant difference was observed between allo-HSCT and ASCT (Figs. 3C–3D).

Figure 3 Survival analysis of AML patients with HSCT.

OS (A) and PFS (B) of AML patients treated with HSCT. (C–D) OS and PFS of AML patients treated with Allo-HSCT and ASCT.

Clinical factors and prognosis

We evaluated whether other risk factors, including age, sex, induction chemotherapy regimen, chromosome karyotype, proportion of BM blasts % (BM%), and proportion of peripheral blood (PB) blasts % (PB%) at initial diagnosis, might affect survival outcomes (Fig. 4). Female sex (OS, 32.4 months vs 15.1 months, P = 0.02; PFS, 17.7 months vs 10.9 months, P = 0.049; Figs. 4A–4B), age < 60 years (OS, 27.9 months vs 10.8 months, P = 0.007; PFS, 17.7 months vs 9.5 months, P = 0.009; Figs. 4C–4D), and a normal karyotype (OS, 38.5 months vs 16.9 months, P = 0.02; PFS, 19.1 months vs 11.7 months, P = 0.15; Figs. 4G–4H) were associated with longer OS and PFS. However, chromosome karyotype seemed to have little effect on PFS (P > 0.05). The chemotherapy regimen was also an important factor, and the IA ± E had longest OS and PFS, the AZA+VEN had the shortest survival (Figs. 4E–4F). BM% ≥ 50% and PB% ≥ 50% were associated with longer OS and PFS (Figs. 5A–5D).

Figure 4 OS and PFS of AML patients based on clinical features.

(A–B) OS and PFS based on gender. (C–D) OS and PFS based on age. (E–F) OS and PFS based on chemotherapy. (G–H) OS and PFS based on karyotypes.

Figure 5 OS and PFS of patients based on BM%, PB% and patients with AML-MR.

(A–B) OS and PFS based on BM%. (C–D) OS and PFS based on PB%. (E–F) OS and PFS based on AML-MR.

Outcomes of patients with AML-RC

Forty-two patients (27.3%) were AML-RC, and 29 patients of these (69.5%) died by the time of follow-up. Median OS and PFS were 16.9 months and 10.6 months, respectively, indicating a prognosis comparable to ASXL1-mutated AML (Figs. 5E–5F).

Prognostic importance of genetic variables for survival

Survival outcomes associated with gene mutations were evaluated (Table 3). TP53 mutations conferred the worst prognosis (median OS: 9.1 months; PFS: 5.3 months), followed by KIT (OS: 13.7 months; PFS: 9.3 months). NPM1 and CEBPA mutations predicted better outcomes (NPM1: OS 38.5 months 26.2 months; CEBPA: OS 27.9 months 17.1 months).

Table 3 Comparison of survival outcomes in patients with AML based on the presence of gene mutations.

Gene	N	Median OS (months)	Median PFS (months)	
FLT3	39	27.9	17.1	
DNMT3A	31	17.9	14.1	
NPM1	30	38.5	26.2	
CEBPA	26	NR	20.7	
TET2	21	22.1	26.1	
ASXL1	20	16.7	10.6	
TP53	20	9.1	5.3	
NRAS	19	26.3	17.6	
WT1	19	NR	10.6	
IDH1	13	NR	NR	
IDH2	14	NR	NR	
KIT	13	13.7	9.3	
Notes.

OS overall survival

PFS progression-free survival

NR not reach

Univariate and multivariate analyses of OS and EFS

The results of the univariate and multivariate analyses of OS and PFS are shown in Figs. S3–S6 and Table 4. In the univariate analysis, age, chemotherapy, HSCT, BM%, and PB% were associated with OS and PFS, while sex and karyotype were also associated with OS but not with PFS. In the multivariate analysis, age, chemotherapy, and HSCT were associated with OS and PFS, whereas BM% and karyotype were also associated with OS but not PFS. Age, chemotherapy (IA ± E), and HSCT were independent predictors of OS and PFS. Notably, age demonstrated conflicting results in univariate vs. multivariate analyses, suggesting confounding by other factors.

Table 4 Univariate and multivariate Cox-proportional hazard regression analyses for OS and PFS.

Variables	OS	PFS	
	HR (95% CI)	P value	HR (95% CI)	P value	
Univariate analysis					
Male gender	1.63 (1.05–2.54)	0.031	1.52 (1.00–2.32)	0.051	
Age ≥ 60	1.88 (1.18–3.00)	0.008	1.79 (1.15–2.79)	0.01	
HB ≥ 70	0.95 (0.60–1.48)	0.808	0.91 (0.59–1.41)	0.684	
PLT ≥ 30	0.95 (0.61–1.48)	0.818	0.77 (0.51–1.17)	0.226	
Normal karyotype	0.61 (0.39–0.95)	0.028	0.73 (0.48–1.12)	0.152	
Normal LDH	0.64 (0.38–1.06)	0.083	0.62 (0.38–1.00)	0.051	
BM (%) ≥50	0.53 (0.34–0.83)	0.005	0.61 (0.40–0.94)	0.023	
PB (%) ≥50	0.59 (0.38–0.93)	0.023	0.64 (0.42–0.98)	0.039	
Chemotherapy					
DA ± E	0.31 (0.17–0.54)	<0.0001	0.29 (0.16–0.50)	<0.0001	
IA ± E	0.16 (0.07–0.40)	<0.0001	0.19 (0.09–0.43)	<0.0001	
Other	0.51 (0.26–1.01)	0.052	0.49 (0.26–0.92)	0.028	
HSCT	0.26 (0.16–0.43)	<0.0001	0.32 (0.21–0.51)	<0.0001	
Multivariate analysis					
Age ≥ 60	0.29 (0.12–0.70)	0.006	0.29 (0.12–0.66)	0.003	
Normal karyotype	0.60 (0.37–0.97)	0.036	0.75 (0.48–1.18)	0.212	
BM (%) ≥50	0.52 (0.30–0.90)	0.018	0.64 (0.38–1.08)	0.096	
PB (%) ≥50	1.06 (0.59–1.93)	0.837	1.03 (0.58–1.84)	0.919	
Chemotherapy					
DA ± E	0.19 (0.06–0.58)	0.004	0.16 (0.05–0.44)	<0.0001	
IA ± E	0.14 (0.04–0.52)	0.004	0.14 (0.04–0.48)	0.002	
Other	0.58 (0.25–1.38)	0.219	0.48 (0.21–1.09)	0.081	
HSCT	0.29 (0.17–0.49)	<0.0001	0.36 (0.22–0.60)	<0.0001	

Discordant outcomes within ELN risk groups

Patients in the 2022 ELN favorable/intermediate (Group A) and adverse (Group B) groups exhibited divergent outcomes. With an OS of 18 months as the criterion, a subgroup analysis was urgently conducted: A (OS > 18 months), A (OS < 18 months), B (OS > 18 months), and B (OS < 18 months). Table S3 shows the patient-related variables in the groups of patients. Subgroup analysis revealed that FLT3, DNMT3A, NPM1, KIT, CEBPA, and TET2 mutations in Group A (Table S4) and ASXL1, CEBPA, TET2, WT1, NRAS, and FLT3 in Group B (Table S5) influenced prognosis. These findings highlight the prognostic impact of specific mutations beyond 2022 ELN classifications.

Discussion

AML prognosis is influenced by genetic and clinical factors, necessitating individualized risk stratification (Venugopal & Sekeres, 2024). To our knowledge, our retrospective study is leveraging real-world data to evaluate the prognostic utility of NGS-based risk classification under the 2022 ELN framework. We investigated the prognostic implications of genetic mutations, clinical features, and treatment modalities under the 2022 ELN framework. Notably, while the 2022 ELN guidelines enhance the accuracy of intermediate/adverse risk stratification, they exhibit limitations in capturing intragroup variability, particularly among patients harboring FLT3, CEBPA, or TET2 mutations. Mutations in TP53 and KIT emerged as strong predictors of dismal outcomes, underscoring their necessity for inclusion in future prognostic models. The 2022 ELN framework demonstrated superior discriminatory power for OS and PFS compared to 2017 ELN, particularly in refining adverse-risk classification (Dohner et al., 2022).

HSCT confers a survival advantage in AML patients, with no significant differences observed between allo-HSCT and ASCT (Chen et al., 2023). At present, there are still controversies regarding the selection criteria and therapeutic efficacy of allo-HSCT and ASCT in AML. Patients with intermediate to high risk may be more likely to benefit from allo-HSCT. However, some studies have found that for low-risk patients such as Core-binding factor acute myeloid leukemia (CBF-AML), the efficacy of ASCT is superior to that of allo-HSCT (Al Hamed et al., 2024). Due to the limitations of our sample size, this conclusion still requires further verification. Our study identified female sex, age < 60 years, and normal karyotype as independent predictors of prolonged OS and PFS. Notably, conflicting results emerged between univariate and multivariate analyses for age, suggesting that its prognostic impact may be modulated by co-factors such as mutation profiles or treatment intensity. Although age remained a significant predictor in the univariate analysis, its marginal effect in multivariate modeling likely reflects the limited representation of elderly patients (15.6%) in our cohort. In addition, our study revealed that BM% ≥50% and PB% ≥50% are not factors for poor prognosis; in contrast, they are factors for good prognosis, which is not quite what we thought. This result needs to be further verified.

According to the Chinese guidelines for diagnosis and treatment of adult AML, for all AML patients who are eligible to participate in clinical studies, it is recommended that they primarily participate in clinical studies. For patients who are suitable for intensive treatment, the IA or DA regimens are the mainstream options. For those who are not suitable for intensive treatment or are elderly and frail, the AZA + VEN regimen is the mainstream option (Leukemia Lymphoma Group CSoHCMA, 2023). There are no uniform OS or PFS values applicable to all Chinese AML patients. This is usually significantly influenced by factors such as age, genetic risk, physical condition and treatment tolerance. One retrospective, real-world study demonstrated that patients with AML treated with VEN+AZA at a single centre had a median OS of 11.93 months (Bai et al., 2025). Our study revealed that chemotherapy prolonged survival, with the IA ± E regimen resulting in the longest OS and PFS and AZA+VEN resulting in the shortest OS and PFS, which is consistent with the findings of one study (Bell et al., 2019). The reason for this result may be that AZA+VEN is mainly used for elderly patients, those with multiple comorbidities or poor physical condition, or those who cannot tolerate intensive chemotherapy. These patients have a poorer prognosis due to their underlying diseases and physical conditions, which limit the intensity and efficacy of the treatment. The AZA+VEN regimen is a low-intensity chemotherapy approach, which is unable to completely eliminate leukemia cells, has a high recurrence rate, and a low long-term survival rate. In addition, HSCT is also an important factor in prolonging the survival prognosis of patients.The 2022 ELN and 2017 ELN are not suitable for elderly patients receiving nonintensive therapy, especially AZA+VEN (Jahn et al., 2023). These patients can be referred to 2022 ELN patients receiving low-intensity therapy (Dohner et al., 2024a). Our results are consistent with reports that the IA regimen prolongs OS and PFS in AML patients more than the DA regimen does (Ding et al., 2024). These results emphasize the need for risk-adapted treatment strategies, particularly in non-intensive settings where 2022 ELN guidelines may not apply.

HSCT significantly improved survival, though its benefit was limited in TP53-mutated patients (He et al., 2024; Kim et al., 2024). Notably, HSCT failed to confer survival benefits in this subgroup, with all five patients undergoing transplantation dying post-transplant (Zhao et al., 2023). OS and PFS were the shortest for TP53, followed by KIT. The occurrence rate of the WT1 gene in our study was 12.34%. TP53 is mutually exclusive with WT1, while CEBPA is prone to coexist with WT1. The median OS of WT1 was not reached, and the median PFS was 10.6 months. In addition, our study found that the CEBPA group indicated a good prognosis, but if combined with WT1 gene mutation, the prognosis might be poor (Wang et al., 2022). KIT mutations, not included in ELN criteria, demonstrated prognostic significance comparable to TP53, necessitating their reevaluation in risk models. Interestingly, we found a strange phenomenon, 2022 ELN risk stratification is the same, but the actual prognosis is completely different. Statistical analysis of their baseline characteristics revealed no significant differences in Group A, except for LDH level and HSCT. In Group B, the only significant difference was PB%. Co-mutation analysis revealed distinct patterns—Group A (OS < 18 months) was enriched for NPM1, FLT3, CEBPA and TET2 co-mutations, whereas Group B (OS > 18 months) exhibited frequent TET2, ASXL1, WT1 and CEBPA associations. These findings highlight the inadequacy of current risk models in accounting for mutation interactions and suggest that 2022 ELN stratification may misclassify patients with specific co-mutation profiles. As proposed by Dohner et al. (2024b), a four-gene model (TP53, FLT3-ITD, NRAS, KRAS) demonstrates superior predictive accuracy for elderly AML patients receiving non-intensive therapy. Our study emphasizes the need for integrated models that incorporate both ELN criteria and mutation-specific signatures. These results advocate for a refined risk model integrating ELN criteria with mutation-specific signatures, particularly for elderly or non-intensive therapy patients. Future studies should validate the prognostic role of KIT and explore gene-environment interactions to optimize personalized care. The retrospective design and single-center cohort limit generalizability. Larger prospective studies are needed to validate prognostic role and optimize co-mutation scoring systems.

Conclusions

In conclusion, AML prognosis is influenced by genetic and clinical factors, necessitating individualized risk stratification. Our study confirms that the 2022 ELN guidelines improve intermediate/adverse risk classification but underestimate variability within groups, particularly in patients with FLT3, CEBPA, or TET2 mutations. TP53 and KIT mutations were strongly associated with poor outcomes, warranting inclusion in prognostic models. HSCT significantly improved survival, though its benefit was limited in TP53-mutated patients. Our findings underscore the need for molecular data integration to refine AML risk stratification.

Supplemental Information

Supplemental Information 1 Supplementary Information

Supplemental Information 2 Raw data

We thank all participating patients and their families.

Additional Information and Declarations

Competing Interests

Author Contributions

Human Ethics

Data Availability

The authors declare there are no competing interests.

Fanqiao Meng conceived and designed the experiments, performed the experiments, analyzed the data, prepared figures and/or tables, authored or reviewed drafts of the article, and approved the final draft.

Maoyuan Xiang performed the experiments, analyzed the data, prepared figures and/or tables, authored or reviewed drafts of the article, and approved the final draft.

Huan Xie conceived and designed the experiments, performed the experiments, analyzed the data, prepared figures and/or tables, authored or reviewed drafts of the article, and approved the final draft.

Yu Liu analyzed the data, prepared figures and/or tables, authored or reviewed drafts of the article, and approved the final draft.

Yan Qi analyzed the data, prepared figures and/or tables, authored or reviewed drafts of the article, and approved the final draft.

Dongfeng Zeng analyzed the data, prepared figures and/or tables, authored or reviewed drafts of the article, and approved the final draft.

The following information was supplied relating to ethical approvals (i.e., approving body and any reference numbers):

The study protocol was conducted following the principles of the Declaration of Helsinki and approved by the Ethics Committee Review Board of the Third Hospital of Army Medical University (2025-09).

The following information was supplied regarding data availability:

The raw data is available in the Supplemental File.

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
