# Peer review of "Retrospective study on the clinical outcomes and characteristics of acute myeloid leukemia: different outcomes in the same risk group"

_PeerJ, doi:10.7717/peerj.20436_

## Round 0.1 · original submission · Major Revisions

· Academic Editor

Major Revisions

·

Basic reporting

1. What is the current prevalence of AML cases among your Chinese population?
2. What are the OS and RFS of AML cases among your Chinese population?
3. What are the current treatment procedures going on in China for AML?
4. In the GLOBOCAN 2022 data, leukemia is most prevalent in China. What methodologies have you adopted to manage AML cases?
5. Define the role of cytogenetics in the diagnosis of AML.

Experimental design

1. How did you calculate the sample size for this study?
2. Is the FAB Classification you have adopted for the inclusion and exclusion of M3-AML Subtype?
3. How have you designed an NGS panel targeting 56 myeloid-related genes?
4. You have not explained the pipeline for how you did the NGS Analysis.
5. Methodology is unclear; cytogenetics, molecular analysis, and fish all these things were missing.

Validity of the findings

1. We also workon this gene WT1 on AML. Which variant of mutation have you found in this gene?
2. During NGS Analysis, you did not find any novel variant.
3. What are 0.1, 0.2, 0.3 on the X axis of Figure 1A, a bar graph showing the frequency of mutations in AML.
4. What are the top oncogenic signaling pathways you got by performing NGS Analysis?

Reviewer 2 ·

Basic reporting

The language must be revised. There are many small errors, some errors that makes the sentence unclear. I have comments don some of the references

Experimental design

ok

Validity of the findings

ok

Annotated reviews are not available for download in order to protect the identity of reviewers who chose to remain anonymous.

---

## Round 0.2 · Major Revisions

· Academic Editor

Major Revisions

Although reviewer 2 has only minor concerns, reviewer 1 notes that the revised manuscript does not incorporate appropriate changes in response to his earlier comments. When submitting your revised manuscript, please include a response to reviewer 1 that explains how your manuscript has been revised. Please also make appropriate changes to address the suggestions of reviewer 2.

·

Basic reporting

My previous review comments were not incorporated into this revised manuscript. I am not satisfied with the current version of the manuscript.

Experimental design

-

Validity of the findings

-

Reviewer 2 ·

Basic reporting

See attached PDF

Experimental design

-

Validity of the findings

-

Annotated reviews are not available for download in order to protect the identity of reviewers who chose to remain anonymous.

---

## Round 0.3 · accepted · Accept

· Academic Editor

Accept

Thank you for revising your manuscript to address the concerns of the reviewers. Reviewer 1 now recommends acceptance and I am satisfied that the comments of reviewer 2 have been addressed. The manuscript is now ready for publication.

·

Basic reporting

ALL THE COMMENTS WERE PROPERLY ADDRESSED BY THE AUTHORS. IT SEEMS FINE NOW.

Experimental design

ALL THE COMMENTS WERE PROPERLY ADDRESSED BY THE AUTHORS. IT SEEMS FINE NOW.

Validity of the findings

ALL THE COMMENTS WERE PROPERLY ADDRESSED BY THE AUTHORS. IT SEEMS FINE NOW.